# Development and validation of the Multimorbidity Treatment Burden Questionnaire (MTBQ)

Polly Duncan ,[1] Mairead Murphy,[1] Mei-See Man ,[2] Katherine Chaplin ,[2] Daisy Gaunt,[3] Chris Salisbury

'Development and validation of the Multimorbidity Treatment Burden Questionnaire' was presented as an oral presentation at the Annual Society for Academic Primary Care Conference, Warwick, UK, 12 July 2017.

[1]Centre for Academic Primary Care, University of Bristol, Bristol, UK
[2]School of Social and Community Medicine, University of Bristol, Bristol, UK
[3]Bristol Randomised Trials Collaboration, School of Social and Community Medicine, Faculty of Medicine and Dentistry, University of Bristol, Bristol, UK

**Correspondence to**
Dr Polly Duncan;
polly.duncan@bristol.ac.uk

## ABSTRACT

**Objective** To develop and validate a new scale to assess treatment burden (the effort of looking after one's health) for patients with multimorbidity.

**Design** Mixed-methods.

**Setting** UK primary care.

**Participants** Content of the Multimorbidity Treatment Burden Questionnaire (MTBQ) was based on a literature review and views from a patient and public involvement group. Face validity was assessed through cognitive interviews. The scale was piloted and the final version was tested in 1546 adults with multimorbidity (mean age 71 years) who took part in the 3D Study, a cluster randomised controlled trial. For each question, we examined the proportion of missing data and the distribution of responses. Factor analysis, Cronbach's alpha, Spearman's rank correlations and longitudinal regression assessed dimensional structure, internal consistency reliability, construct validity and responsiveness, respectively. We assessed interpretability by grouping the global MTBQ scores into zero and tertiles (>0) and comparing participant characteristics across these categories.

**Results** Cognitive interviews found good acceptability and content validity. Factor analysis supported a one-factor solution. Cronbach's alpha was 0.83, indicating internal consistency reliability. The MTBQ score had a positive association with a comparator treatment burden scale ($r_s$ 0.58, P<0.0001) and with self-reported disease burden ($r_s$ 0.43, P<0.0001), and a negative association with quality of life ($r_s$ −0.36, P<0.0001) and self-rated health ($r_s$ −0.36, P<0.0001). Female participants, younger participants and participants with mental health conditions were more likely to have high treatment burden scores. Changes in MTBQ score over 9-month follow-up were associated, as expected, with changes in measures of quality of life (EuroQol five dimensions, five level questionnaire) and patient-centred care (Patient Assessment of Chronic Illness Care).

**Conclusion** The MTBQ is a 10-item measure of treatment burden for patients with multimorbidity that has demonstrated good content validity, construct validity, reliability and responsiveness. It is a useful research tool for assessing the impact of interventions on treatment burden.

**Trial registration number** ISRCTN06180958.

## Strengths and limitations of this study

► A concise, simply worded measure based on an evidence-based framework to include all the important aspects of treatment burden was developed and validated.
► The measure was comprehensively tested using international standards for validating questionnaires.
► The measure was validated in 1546 mostly elderly patients with three or more long-term conditions.
► Study participants were recruited into a trial, which may limit generalisability.
► High floor effects were found similar to other existing treatment burden questionnaires.

medical conditions and the impact that this has on their general well-being.[1] This includes complex medication regimens, coordinating healthcare appointments, making lifestyle changes and self-monitoring.

This is particularly relevant to patients with multimorbidity (having multiple long-term conditions). Associated with the ageing population, multimorbidity has become the norm, affecting over two-thirds of adults attending general practice.[2] Current health policy envisages greater support for patients to self-manage their chronic medical conditions. However, the time and energy this requires of patients can be overwhelming.[3]

In order to understand the impact of treatment burden, and particularly to assess the effects of interventions that might increase or decrease burden, a valid patient-reported outcome measure (PROM) is essential. There are four existing PROMs that measure aspects of treatment burden for patients with multimorbidity,[4–8] all of which have important limitations. The 13-question Treatment Burden Questionnaire (TBQ) was originally developed in French, and subsequently a revised 15-question English version was tested.[4 5] Some of the content is healthcare system-specific and the wording

## INTRODUCTION

Treatment burden is a patient's perception of the effort required to self-manage their

is relatively complex, perhaps reflecting the fact that the English version was tested in a relatively young and highly educated population of volunteers recruited from the 'Patients like me' website (mean age 51 years, 78% with college education), not all of whom had multimorbidity.[4] The Patient Experience with Treatment and Self-management (PETS) PROM was recently developed in the USA and includes 48 questions grouped under nine separate domains of treatment burden.[8] While this measure is comprehensive, its length is a limitation. The Multimorbidity Illness Perceptions Scale (MULTIPLes) was developed and validated in elderly patients (mean age 70 years) with multimorbidity and includes a six-question treatment burden subscale and a three-question activity limitation subscale.[7] This measure is brief but omits several important aspects of treatment burden. Similarly, the 11-question Healthcare Task Difficulty (HCTD) questionnaire was designed to measure only one aspect of treatment burden.[6]

The purpose of this study was to develop and validate a new concise measure of treatment burden for patients with multimorbidity.

## METHODS
### Study setting
This questionnaire was developed and validated as part of the 3D Study, a multicentre, cluster randomised control trial that aims to improve the management of patients with multimorbidity within primary care.[9] Participants aged 18 years or older with three or more of the long-term conditions included in the 2014 UK Quality and Outcomes Framework were recruited from 33 general practices in three areas of the UK.

### Development of the questionnaire
We reviewed the literature on the concept and measurement of treatment burden in multimorbidity using PubMed in July 2014. We identified a number of relevant qualitative studies[10–12] and three relevant existing PROMs that were not specific to a particular medical condition. These were the TBQ,[4 5] the MULTIPLes[7] and the HCTD questionnaires.[6] A further measure, the PETS scale, was published later.[8] We identified relevant domains for the PROM by reviewing the three existing PROMs against a framework of treatment burden which had been developed following qualitative interviews and focus groups.[1] We then sought the views from a patient and public involvement (PPI) group of eight patients with multimorbidity formed for the purpose of the 3D Study, discussing the concept of treatment burden, the existing measures, the treatment burden framework and the domains of treatment burden to be included in the questionnaire. We then developed a draft questionnaire with 12 questions and undertook two rounds of cognitive interviews with eight PPI group members to improve the face and content validity of the scale (online supplementary appendix A).[13] Participants were asked to 'think aloud'[13] as they completed the questionnaire commenting on the reasoning behind their ratings; and perceived question meaning, the layout, title, introduction and general wording. They also gave their own examples of treatment burden and reflected on whether these would be captured by the questionnaire. Modifications to the questionnaire were made between the two rounds and an additional question was added about accessing healthcare during the evenings and weekends (see the Results section). Following written consent, the interviews were audio-taped and field notes were taken. The second round of cognitive interviews led to only minor changes to the questionnaire with no new insights emerging. A debriefing meeting was held with PPI members and final changes to the 13-question questionnaire were made.

### Recruitment, data collection and measures
Data were collected in two related studies, the cross-sectional 3D pilot study and the longitudinal main 3D Study, a cluster randomised controlled trial. The 13 candidate questions were included in a questionnaire, named the Multimorbidity Treatment Burden Questionnaire (MTBQ). Sociodemographic information (see table 1) was collected at baseline in both the pilot and main studies. Details of participants' medical conditions were collected from their family practice computer records. Measures of health-related quality of life (EQ-5D-5L),[14] self-rated health (single question item), self-reported disease burden (Bayliss)[15] and patient-centred care (Patient Assessment of Chronic Illness Care (PACIC))[16] were collected at baseline and 9 months in both the pilot and main 3D studies. Following a review of existing measures and discussion with the PPI group, the HCTD[6] questionnaire was included in the pilot study questionnaire as the best comparator for the MTBQ. A key reason for choosing this measure was the simple wording and brevity. This was felt to be important because many of the participants of the 3D Study were older people and some had low literacy levels.

The questionnaire was sent to participants by post. For non-responders, a reminder letter was sent 10–14 days later, and a second reminder phone call was made 10–14 days after this.

### Analysis
Data were analysed using STATA V.14. We generated descriptive statistics of participant characteristics for the pilot and main studies. The pilot study data were used to test the prespecified hypothesis of a positive association between global MTBQ score and HCTD score. The main study data were used for the remainder of the analysis.

We tested the psychometric properties of the questionnaire against the minimum standards set out by the International Society for Quality of Life Research (ISOQOL).[17] The analysis plan and results are described in relation to ISOQOL's six recommended standards.

## 1. Conceptual and measurement model
### Conceptual framework
See the 'Development of the questionnaire' section.

### Question properties
To assess the properties of the questions, we examined the proportion of missing data and 'does not apply' responses and the distribution of responses. Responses of 'not difficult' or 'does not apply' were scored as 0. Floor and ceiling effects of the MTBQ were compared with the HCTD.[6] Questions with a proportion of 'does not apply' responses greater than 40% were removed and excluded from the analysis.

### Dimensionality
To examine the dimensionality of the scale, we performed factor analysis. This is a statistical technique used to reduce a larger number of items into a smaller number of common factors that reflect shared variance.[18] Items that share a lot of variance should have high 'loadings' (correlation between the item and the factor) and low uniqueness (variance that is unique to the item, not common to the factor). Loading of at least 0.4 and uniqueness of less than 0.6 are acceptable.[19] The number of factors extracted was decided by a combination of Kaiser's rule (eigenvalues greater than 1),[20] the scree plot[18] and by interpretability of domains.

## 2. Reliability
To test internal consistency reliability, we examined the interitem correlation matrix and calculated Cronbach's alpha, a measure of consistency between the items in a scale. Interitem correlations between 0.2 and 0.4 were deemed ideal.[21] A Cronbach's alpha of 0.7–0.9 was acceptable.[22]

## 3. Validity
### Content validity
The content validity of the questionnaire was tested iteratively using cognitive interviews (see the 'Development of the questionnaire' section).

### Construct validity
Each question was scored as follows: 0 (not difficult/does not apply), 1 (a little difficult), 2 (quite difficult), 3 (very difficult) and 4 (extremely difficult). Participants were excluded if more than 50% of their responses were missing. To calculate a global score, each participant's average score was calculated from the questions answered and multiplied by 25 to give a score from 0 to 100.

Construct validity was examined by testing five prespecified hypotheses: first, a positive association between global MTBQ score and global HCTD score[6]; second, a negative association between global MTBQ score and health-related quality of life (EQ-5D-5L)[14]; third, a positive association between global MTBQ score and self-reported disease burden score[15]; fourth, a positive association between global MTBQ score and number of self-reported comorbidities[15]; and fifth, a negative association between global MTBQ and self-rated health (single question item). We applied Spearman's rank correlation to test these hypotheses.

### Responsiveness
According to the ISOQOL guidelines, responsiveness to change should be assessed.[17] Due to the non-normal distribution of the global MTBQ score, standard methods to assess responsiveness to change such as calculating an effect size[22] were not possible. We therefore tested the responsiveness of the global MTBQ score by assessing whether changes over time in measures of quality of life (EQ-5D-5L)[14] and patient-centred care (PACIC)[16] were inversely associated with changes in MTBQ as anticipated. We used a linear regression model of the standardised change in quality of life (EQ-5D-5L) score between baseline and 9 months on the standardised change in MTBQ between baseline and 9 months. These standardised change scores were calculated at the participant level by dividing the individual difference in 9-month and baseline MTBQ (or EQ-5D-5L) score by the SD of the overall MTBQ (or EQ-5D-5L) change score for all individuals. We then further adjusted this linear regression model in a subsequent analysis by age, gender, number of long-term conditions and individual participant deprivation level. All participants who died prior to the 9-month follow-up were given an EQ-5D-5L follow-up score of 0.

We then used the same model for MTBQ specified as above but included the standardised change in PACIC scores between baseline and 9-month follow-up, defined as previously, and subsequently further adjusted this model by the additional covariates as specified.

## 4. Interpretability of scores
The distribution of global MTBQ scores was examined and compared with the distribution of HCTD[6] scores.

We assessed interpretability of the questionnaire by grouping the global MTBQ scores greater than 0 into tertiles. Four categories were generated: no burden (score 0), low burden (score <10), medium burden (10–22) and high burden (≥22). Participant characteristics and key outcome variables, including EQ-5D-5L,[14] Bayliss disease burden score[15] and self-rated health, were compared across these four categories. To test for associations between treatment burden score category and participant characteristics, we performed ordinal logistic regression of MTBQ group (four treatment burden categories) on each participant characteristic. We then further adjusted these ordinal logistic regression models by age, gender, number of comorbidities, age left full-time education and individual deprivation score.

## 5. Translation
Not applicable.

## 6. Demands on patient respondents and investigators
The effort required of patient respondents to complete the questionnaire was assessed during the cognitive interviews, and by reviewing the proportion of missing

**Table 1**  Participant characteristics (main study n=1546, pilot study n=143)

| | Pilot study n/N* (%) | Main study n/N* (%) |
|---|---|---|
| Mean age (SD) | 74 (10) | 71 (12) |
| Age (years) | | |
| ≤50 | 3 (2) | 79 (5) |
| 51–60 | 9 (6) | 196 (13) |
| 61–70 | 27 (19) | 420 (27) |
| 71–80 | 67 (47) | 510 (33) |
| 81–90 | 33 (23) | 315 (20) |
| ≥90 | 4 (3) | 26 (2) |
| Gender | | |
| Male | 65 (45) | 763 (49) |
| Number of comorbidities | | |
| 3 | 109 (76) | 1234 (80) |
| 4 | 23 (16) | 277 (18) |
| 5 | 10 (7) | 31 (2) |
| 6 | 1 (<1) | 4 (<1) |
| Comorbidities* | | |
| Cardiovascular disease | 138 (97) | 1445 (97) |
| Stroke/Transient ischaemic attack | 35 (25) | 527 (34) |
| Diabetes | 63 (44) | 811 (52) |
| Chronic kidney disease | 83 (58) | 464 (30) |
| Chronic obstructive pulmonary disease or asthma | 58 (41) | 770 (50) |
| Epilepsy | 6 (4) | 76 (5) |
| Atrial fibrillation | 46 (32) | 529 (34) |
| Severe mental health problems† | 2 (1) | 66 (4) |
| Depression | 26 (18) | 560 (36) |
| Dementia | 6 (4) | 60 (4) |
| Learning disability | 3 (2) | 14 (1) |
| Rheumatoid arthritis | 9 (6) | 103 (7) |
| Heart failure | 14 (10) | 157 (10) |
| Ethnicity | | |
| White British | 135/136 (99) | 1502/1519 (99) |
| Age left full-time education (years) | | |
| ≤14 | 22 (15) | 154/1541 (10) |
| 15 or 16 | 74 (52) | 907/1541 (59) |
| 17 or 18 | 25 (17) | 222/1541 (14) |
| ≥19 | 22 (15) | 258/1541 (17) |
| Employment status | | |
| Fully retired from work | 113/139 (81) | 1044/1501 (70) |

Continued

**Table 1**  Continued

| | Pilot study n/N* (%) | Main study n/N* (%) |
|---|---|---|
| Deprivation score quartile‡ | | |
| England | | |
| Lower quartile | 99/143 (69) | 445/1079 (41) |
| Middle lower quartile | 44/143 (31) | 304/1079 (28) |
| Middle upper quartile | 0 | 196/1079 (18) |
| Upper quartile | 0 | 134/1079 (12) |
| Scotland | | |
| Lower quartile | | 105/467 (22) |
| Middle lower quartile | | 46/467 (10) |
| Middle upper quartile | | 156/467 (33) |
| Upper quartile | | 160/467 (34) |
| Baseline scores of outcome measures | | |
| Mean HCTD score§ (SD, N) | 1.14 (1.7, 143) | |
| Mean self-reported disease burden score¶(SD, N) | | 19 (12.4, 1458) |
| Mean number of self-reported conditions** (SD, N) | | 8 (3.2, 1543) |
| Mean quality of life score†† (SD, N) | | 0.6 (0.3, 1542) |
| Mean self-rated health score‡‡ (SD, N) | | 2 (0.8, 1523) |
| Mean patient-centred health score§§(SD, N) | | 2.5 (1.0, 1232) |

*For characteristics where there are no missing data, n is shown; for characteristics with missing data n/N is shown.
†Including schizophrenia and psychotic illness.
‡Individual Index of Multiple Deprivation score, 2010, for England, and Scottish Index of Multiple Deprivation score, 2012, for Scotland, based on participants' postcodes. The lower quartile is the least deprived and the upper quartile is the most deprived.
§Calculation of global HCTD score: sum of scores where each question was scored 0 (no difficulty), 1 (some difficulty) or 2 (a lot of difficulty). Minimum score 0, maximum score 16. Missing data were scored 0 (not difficult), as suggested by the HCTD authors.[6]
¶Sum of the weighted scores (each scored 1–5) from the Bayliss scale.[12] Responses were excluded if participants ticked that they had a condition but did not score how much the condition limited their daily activity, or if they gave a score without ticking that they had the condition.
**Number of self-reported conditions from a list of 27 conditions itemised in the Bayliss scale.
††EuroQol five dimensions, five level questionnaire (EQ-5D-5L) score.[11]
‡‡Single question: 'In general, would you say your health is poor (1), fair (2), good (3), very good (4) or excellent (5)?'
§§Patient Assessment of Chronic Illness Care score.[13]
HCTD, Healthcare Task Difficulty.

responses. We set out to reduce the demands on investigators by providing clear instructions on how to calculate a global MTBQ score, including handling of missing data, and how to report and interpret these scores.

### Ethical approval and data sharing

The study was registered under trial registration number ISRCTN06180958 (main trial results yet to be published). Data will be available from the University of Bristol Research Data Storage Facility after the main results of the 3D trial have been published in 2018.

## RESULTS
### Participant characteristics

One hundred and forty-three adults participated in the pilot study. From 1546 participants in the main 3D Study who completed the main baseline questionnaire, we were able to calculate an MTBQ score for 1524 (99%) individuals who completed at least half of the baseline MTBQ questions. At 9-month follow-up, 1356 returned the questionnaire and an MTBQ score could be calculated for 1299 (96%). The participants were mostly elderly (mean age 71 years for the main study), fully retired from work, had left school aged 16 years or younger, and 99% were white British (table 1). Around two-thirds of participants from England lived in areas of low deprivation (low or middle lower quartiles), whereas almost two-thirds of participants from Scotland lived in areas of high deprivation (middle upper or upper quartiles).

### Conceptual and measurement model
#### Conceptual framework

The framework developed by Eton et al[1] describes three major themes of treatment burden: the work required to look after one's health (eg, self-monitoring, making lifestyle changes); tools and strategies patients use to reduce their treatment burden (eg, organising medication); and factors that increase burden (eg, poor continuity of care). We mapped the three existing treatment burden questionnaires against this framework, and discussed this with the PPI group who felt that all of the domains of treatment burden identified in the literature should be included in the PROM. We had initially considered excluding questions about costs since healthcare is mostly free under the National Health Service, but our PPI group argued that they still experienced additional costs from managing illness so this domain was retained in the first draft.

#### Question properties

The proportion of missing data for each question was between 1% and 3% (see table 2). Questions 3, 9 and 10 with a high proportion of 'does not apply' responses (table 2) were excluded from the main analysis. Since these questions might apply to other populations, we repeated Cronbach's alpha including these questions in the various combinations (online supplementary appendix B). These extra questions may be considered as

optional depending on the study population. Responses were positively skewed and a floor effect was found for some questions. However, the MTBQ had fewer floor effects than the comparator HCTD (online supplementary appendix C).

The global MTBQ scores were also skewed with 26% of pilot study participants and 22% of main study participants scoring 0 (online supplementary appendix D). Again, the HCTD had greater floor effects, with 54% of participants having a global score of 0.

#### Dimensionality

Both Kaiser's 'eigenvalue greater than one' rule and Cattell's scree plot criterion suggested a one-factor solution, and this explained 93% of the common variance. Loadings on this factor were uniformly greater than 0.4. The factor solution had high uniqueness for some items. This can sometimes indicate that the item is not strongly related to others,[18] but because of the important content of these variables (eg, lifestyle changes, collecting medication), we chose to include them.

#### Reliability

Questions 1 and 2 have a high interitem correlation of 0.69, and questions 6 and 7 have an interitem correlation of 0.62 (online supplementary appendix E). Almost all of the other interitem correlations were in the ideal range of 0.2–0.4. A decision was made to include questions 1 and 2, and 6 and 7, despite the high interitem correlation coefficients because it was felt these questions were about different aspects of treatment burden. Cronbach's alpha was 0.83, indicating a high level of internal reliability. Including the optional questions (questions 3, 9 and 10) in various combinations, Cronbach's alpha ranged from 0.82 to 0.84, again demonstrating good internal consistency (see online supplementary appendix B).

#### Validity
#### Face and content validity

Participants from the PPI group commented that the wording was clear and easy to understand. One participant felt that accessing healthcare outside of usual general practitioner (GP) opening hours caused significant treatment burden for him. In response to this, we added a question about difficulty getting healthcare in the evenings and weekends (question 10). The remaining participants commented that the important areas of treatment burden were covered by the questionnaire.

#### Construct validity

As predicted, the global MTBQ score had a positive association with the comparator HCTD scale[6] ($r_s$ 0.58, P<0.0001), the Bayliss disease burden scale[12] ($r_s$ 0.43, P<0.0001) and the number of self-reported comorbidities ($r_s$ 0.32, P<0.0001), and a negative association with the quality of life scale[14] ($r_s$ −0.36, P<0.0001) and self-rated health ($r_s$ −0.36, P<0.0001) (table 3). This provides good evidence on the construct validity of the scale.

**Table 2** Responses to the Multimorbidity Treatment Burden Questionnaire (main study baseline data, n=1546)

| Please tell us how much difficulty you have with the following: | N | Not difficult n (n/N %) | A little difficult n (n/N %) | Quite difficult n (n/N %) | Very difficult n (n/N %) | Extremely difficult n (n/N %) | Does not apply n (n/N %) |
|---|---|---|---|---|---|---|---|
| 1. Taking lots of medications | 1518 | 1083 (71) | 257 (17) | 104 (7) | 25 (2) | 20 (1) | 29 (2) |
| 2. Remembering how and when to take medication | 1519 | 1123 (74) | 271 (18) | 60 (4) | 21 (1) | 23 (2) | 21 (1) |
| *3. Paying for prescriptions, over the counter medication or equipment* | 1506 | 312 (21) | 17 (1) | 18 (1) | 4 (<1) | 8 (1) | 1147 (76) |
| 4. Collecting prescription medication | 1514 | 951 (63) | 221 (15) | 63 (4) | 22 (1) | 28 (2) | 229 (15) |
| 5. Monitoring your medical conditions (eg, checking your blood pressure or blood sugar, monitoring your symptoms, etc) | 1513 | 748 (49) | 191 (13) | 111 (7) | 35 (2) | 37 (2) | 391 (26) |
| 6. Arranging appointments with health professionals | 1507 | 765 (51) | 321 (21) | 210 (14) | 81 (5) | 66 (4) | 64 (4) |
| 7. Seeing lots of different health professionals | 1506 | 642 (43) | 309 (21) | 192 (13) | 85 (6) | 68 (5) | 210 (14) |
| 8. Attending appointments with health professionals (eg, getting time off work, arranging transport, etc) | 1512 | 771 (51) | 187 (12) | 107 (7) | 51 (3) | 44 (3) | 352 (23) |
| *9. Getting health care in the evenings and at weekends* | 1496 | 311 (21) | 156 (10) | 184 (12) | 106 (7) | 121 (8) | 618 (41) |
| *10. Getting help from community services (eg, physiotherapy, district nurses, etc)* | 1500 | 393 (26) | 138 (9) | 111 (7) | 51 (3) | 54 (4) | 753 (50) |
| 11. Obtaining clear and up-to-date information about your condition | 1499 | 794 (53) | 263 (18) | 179 (12) | 62 (4) | 47 (3) | 154 (10) |
| 12. Making recommended lifestyle changes (eg, diet and exercise) | 1505 | 534 (35) | 327 (21) | 203 (13) | 112 (7) | 75 (5) | 254 (17) |
| 13. Having to rely on help from family and friends | 1509 | 675 (45) | 213 (14) | 140 (9) | 59 (4) | 70 (5) | 352 (23) |

Notes: Questions 3, 9 and 10 were excluded from the main analysis due to a high proportion of 'does not apply' responses. They are shown in italics. As they may be relevant to other populations, they can be considered as optional.

### Responsiveness

Regression analysis found that for every 1 SD (ie, 0.17) increase in EQ-5D-5L score[14] between baseline and 9-month follow-up, the MTBQ score at follow-up was reduced by 1.7 (regression coefficient −0.14 multiplied by an SD change in MTBQ score of 11.9 (95% CI for regression coefficient −0.19 to −0.08), P<0.0001) (see table 4). This association was also seen after further adjusting the model for the specified covariates (regression coefficient −0.14 (95% CI −0.20 to −0.08), P<0.0001).

The equivalent model for PACIC score[16] showed that for every 1 SD (ie, 0.86) increase in PACIC score between baseline and 9-month follow-up, MTBQ at follow-up was reduced by 1.9 (regression coefficient −0.16 multiplied by an SD change in MTBQ score of 11.9 (95% CI for regression coefficient −0.22 to −0.10), P<0.0001). A similar decrease was also seen after further adjusting the model for the specified covariates (regression coefficient −0.17 (95% CI −0.23 to −0.11), P<0.0001).

### Interpretability of scores

Comparing participants across the four treatment burden groups (no burden, low burden, medium burden and high burden), female participants; younger participants; those with a greater number of long-term conditions; participants with depression, dementia and severe mental health problems listed on their GP records; and participants with worse EQ-5D-5L scores,[14] high disease burden scores[12] and poor self-rated health were more likely to have a high treatment burden score, after adjusting for age, gender, number of comorbidities, age left full-time education and individual deprivation level (see table 5). There was no convincing association between deprivation score and treatment burden score.

### Translation

Not applicable.

### Demands on patient respondents and investigators

We have reduced the effort required from patient responders to complete the questionnaire by developing a short 10-item questionnaire with simple wording, fitting on one side of A4 paper in size 14 font. Participants who took part in the cognitive interviews found this relatively simple to complete, and the proportion of missing data was between 1% and

**Table 3** Association between global MTBQ score and global HCTD score, self-reported disease burden score, quality of life score, number of self-reported conditions and self-rated health at baseline

| Variable | N | Spearman's rank correlations (r$_s$) | P values |
|---|---|---|---|
| Global HCTD score* | 141 | 0.58 | <0.0001 |
| Self-reported disease burden score† | 1443 | 0.42 | <0.0001 |
| Number of self-reported conditions‡ | 1523 | 0.31 | <0.0001 |
| Quality of life score§ | 1520 | −0.36 | <0.0001 |
| Self-rated health¶ | 1503 | −0.36 | <0.0001 |

*Calculation of global HCTD score: sum of scores where each question was scored 0 (no difficulty), 1 (some difficulty) or 2 (a lot of difficulty). Minimum score 0, maximum score 16. Missing data were scored 0 (not difficult), as suggested by the HCTD authors.[6]
†Sum of the weighted scores (each scored 1–5) from the Bayliss scale.[12] Responses were excluded if participants ticked that they had a condition but did not score how much the condition limited their daily activity, or if they gave a score without ticking that they had the condition.
‡Number of self-reported conditions from the Bayliss scale.
§EuroQol five dimensions, five level questionnaire (EQ-5D-5L) score.[11]
¶Single question: 'In general, would you say your health is poor (1), fair (2), good (3), very good (4) or excellent (5)?'
HCTD, Healthcare Task Difficulty; MTBQ, Multimorbidity Treatment Burden Questionnaire.

3%. To reduce demands on investigators, we have provided clear instructions on calculating, reporting and interpreting global MTBQ scores.

## DISCUSSION

In this study, we have developed and validated a 10-item questionnaire, named the Multimorbidity Treatment Burden Questionnaire (MTBQ). The psychometric properties of the questionnaire meet the minimum standards for a PROM set out by ISOQOL,[17] demonstrating good content validity, internal reliability consistency, construct validity and responsiveness. Three additional questions, including one question about the cost of treatment, had a high proportion of 'does not apply' responses in this study population and were omitted from the main analysis. However, these questions may be relevant to other populations (eg, countries where patients pay for prescriptions and healthcare), and the scale remained internally consistent and reliable when they were included, so they may be considered as optional.

We found that younger patients were more likely to report high treatment burden scores and, interestingly, Tran's TBQ found the same phenomenon.[5] There are several possible explanations for this. First, treatment burden may impact more on younger patients because they must juggle their appointments or complex medication regimens alongside having to work or look after dependants. Second, younger patients may have different expectations of how looking after one's health might impact on their lives and, hence, suffer from a greater perceived treatment burden. As expected, we found that patients with mental health conditions including depression and dementia were more likely to have high treatment burden scores. Previous studies have reported similar findings.[6 7] High treatment burden was also associated with having a greater number of long-term conditions. No individual physical condition was found to be associated

**Table 4** Association between global MTBQ score and (1) quality of life (EQ-5D-5L)[11] score and (2) PACIC[13] score

| Outcome | N* | Linear regression coefficient of MTBQ standardised change score (95% CI) | P value | N | Adjusted† linear regression coefficient of MTBQ standardised change score (95% CI) | P value |
|---|---|---|---|---|---|---|
| EQ-5D-5L standardised change score | 1270 | −0.14 (−0.19 to −0.08) | <0.0001 | 1239 | −0.14 (−0.20 to −0.08) | <0.0001 |
| PACIC standardised change score | 930 | −0.16 (−0.22 to −0.10) | <0.0001 | 914 | −0.17 (−0.23 to −0.11) | <0.0001 |
| Outcome | N‡ | SD change in score between baseline and 9-month follow-up | | | | |
| EQ-5D-5L | 1344 | 0.17 | | | | |
| PACIC | 946 | 0.86 | | | | |
| MTBQ | 1285 | 11.9 | | | | |

Results were from linear regression model of standardised change.
*This analysis included participants who completed the outcome questionnaire (EQ-5D-5L or PACIC) and the MTBQ questionnaire at baseline and 9-month follow-up.
†Linear regression model further adjusted for age, gender, number of comorbidities, age left full-time education and individual deprivation score.
‡This analysis included participants who completed the outcome questionnaire (EuroQol five dimensions, five level questionnaire (EQ-5D-5L), PACIC or MTBQ) at baseline and 9-month follow-up.
MTBQ, Multimorbidity Treatment Burden Questionnaire; PACIC, Patient Assessment of Chronic Illness Care.

**Table 5**  Characteristics by categories of treatment burden (main study baseline data)

| | N | None (0) | Low (<10) | Medium (10–22) | High (≥22) | Unadjusted OR* | Adjusted OR† | P value |
|---|---|---|---|---|---|---|---|---|
| Participants | 1524 | 308 | 385 | 425 | 406 | | | |
| Age (mean) | 1524 | 74 | 73 | 71 | 66 | **0.96 (0.95 to 0.97)** | **0.96 (0.95 to 0.97)** | **<0.0001** |
| Gender (n, (%)) | | | | | | | | |
| Male | 651 | 168 (22) | 208 (28) | 193 (26) | 182 (24) | **0.74 (0.62 to 0.88)** | **0.73 (0.60 to 0.87)** | **0.001** |
| Number of long-term conditions (n,(%)) | | | | | | | | |
| 3 | 1217 | 246 (20) | 323 (27) | 335 (28) | 313 (26) | | | |
| 4 or more | 307 | 62 (20) | 62 (20) | 90 (29) | 93 (30) | 1.21 (0.97 to 1.52) | **1.38 (1.09 to 1.74)** | **0.007** |
| Long-term conditions (n, (%)) | | | | | | | | |
| Cardiovascular disease | 1423 | 294 (21) | 367 (26) | 389 (27) | 373 (26) | **0.62 (0.44 to 0.91)** | 0.79 (0.54 to 1.14) | 0.208 |
| Stroke/Transient ischaemic attack | 517 | 127 (25) | 140 (27) | 135 (26) | 115 (22) | **0.69 (0.57 to 0.83)** | 0.82 (0.67 to 1.01) | 0.059 |
| Diabetes | 800 | 158 (20) | 200 (25) | 211 (26) | 231 (29) | 1.13 (0.94 to 1.35) | 1.04 (0.87 to 1.26) | 0.633 |
| Chronic kidney disease | 454 | 101 (22) | 121 (27) | 115 (25) | 117 (26) | 0.86 (0.71 to 1.05) | 1.10 (0.89 to 1.36) | 0.356 |
| Chronic obstructive pulmonary disease or asthma | 758 | 148 (20) | 185 (24) | 222 (29) | 203 (27) | 1.08 (0.90 to 1.29) | 0.91 (0.75 to 1.10) | 0.326 |
| Epilepsy | 76 | 14 (18) | 21 (28) | 24 (32) | 17 (22) | 0.94 (0.63 to 1.41) | 0.76 (0.50 to 1.17) | 0.216 |
| Atrial fibrillation | 524 | 119 (23) | 155 (30) | 142 (27) | 108 (21) | 0.68 (0.56 to 0.82) | 0.91 (0.74 to 1.12) | 0.369 |
| Severe mental health problems‡ | 66 | 7 (11) | 10 (15) | 17 (26) | 32 (48) | **2.61 (1.64 to 4.15)** | **1.75 (1.08 to 2.82)** | **0.022** |
| Depression | 553 | 85 (15) | 105 (19) | 169 (31) | 194 (35) | **1.92 (1.59 to 2.32)** | **1.43 (1.16 to 1.77)** | **0.001** |
| Dementia | 58 | 14 (24) | 10 (17) | 12 (21) | 22 (38) | 1.27 (0.78 to 2.11) | **2.26 (1.34 to 3.81)** | **0.002** |
| Learning disability | 14 | 2 (14) | 2 (14) | 6 (43) | 4 (29) | 1.47 (0.59 to 3.69) | 1.07 (0.36 to 3.21) | 0.907 |
| Rheumatoid arthritis | 102 | 15 (15) | 18 (18) | 40 (39) | 29 (28) | 1.41 (0.99 to 2.01) | 1.28 (0.88 to 1.82) | 0.202 |
| Heart failure | 154 | 36 (23) | 41 (27) | 38 (25) | 39 (25) | 0.85 (0.63 to 1.14) | 1.06 (0.77 to 1.44) | 0.340 |
| Age left full-time education (n, (%))§ | | | | | | | | |
| ≤16 years | 681 | 164 (24) | 172 (25) | 177 (26) | 168 (25) | 1.00 (0.99 to 1.01) | 1.01 (0.99 to 1.02) | 0.450 |
| England | 1078 | 15 | 15 | 15 | 16 | 1.01 (1.00 to 1.01) | 1.00 (0.99 to 1.01) | 0.904 |
| Scotland | 467 | 26 | 26 | 24 | 24 | 1.00 (0.99 to 1.01) | **0.99 (0.99 to 1.00)** | **0.032** |
| EuroQol five dimensions, five level questionnaire (EQ-5D-5L)[11] (mean) | 1520 | 0.67 | 0.63 | 0.56 | 0.42 | **0.11 (0.08 to 0.16)** | **0.09 (0.06 to 0.12)** | **<0.0001** |
| Disease burden score[12] (mean) | 1443 | 12.8 | 15.7 | 19.0 | 26.1 | **1.06 (1.06 to 1.08)** | **1.07 (1.07 to 1.09)** | **<0.0001** |
| Self-rated health (n, (%)) | | | | | | | | |
| Poor | 315 | 36 (11) | 42 (13) | 75 (24) | 162 (51) | | | |
| Fair | 674 | 112 (17) | 168 (25) | 216 (32) | 178 (26) | **0.39 (0.30 to 0.50)** | **0.41 (0.31 to 0.53)** | **<0.0001** |
| Good | 422 | 111 (26) | 138 (33) | 116 (27) | 57 (14) | **0.20 (0.15 to 0.26)** | **0.19 (0.14 to 0.26)** | **<0.0001** |
| Very good | 87 | 40 (46) | 28 (32) | 16 (18) | 3 (3) | **0.08 (0.05 to 0.13)** | **0.08 (0.05 to 0.12)** | **<0.0001** |
| Excellent | 5 | 3 (60) | 2 (40) | 0 | 0 | **0.04 (0.01 to 0.23)** | **0.03 (0.00 to 0.16)** | **<0.0001** |

*Ordinal logistic regression comparing no burden (0), low burden (<10), medium burden (10–22) and high burden (≥22).
†Ordinal logistic regression comparing no burden (0), low burden (<10), medium burden (10–22) and high burden (≥22), adjusted for age, gender, number of comorbidities, age left full-time education and individual deprivation score.
‡Including schizophrenia and psychotic illness.
§Individual Index of Multiple Deprivation score, 2010, for England, and Scottish Index of Multiple Deprivation score, 2010, for Scotland, for both a higher score correlates with greater deprivation.
Statistically significant associations are shown in bold

with high treatment burden. This result differs from both the TBQ study, which found an association between treatment burden and diabetes, and the HCTD study, which found an association between treatment burden and stroke, congestive heart failure and falls.[5][6] As expected, participants with low quality of life (EQ-5D-5L)[14] score, high disease burden score[15] and poor self-rated health were more likely to have high treatment burden. We also found that female participants were more likely to report high treatment burden compared with male participants. This has not been reported elsewhere. There was no association between deprivation level and treatment burden score. One might expect that people from more deprived areas might have fewer support networks and resources and so would experience higher treatment burden. Alternatively, one could argue that participants from more deprived areas might be more accepting of how looking after their health impacts on their day-to-day life and so report lower treatment burden.

A key strength of this study is that the MTBQ has been validated in a large sample of participants for whom it is intended—elderly multimorbid patients with a mean age of 71 years and three or more long-term conditions. In comparison, the English version of the Tran's TBQ was validated in a younger computer-literate population with a mean age of 51 years.[4][5] The MTBQ had good face validity, was found to be user-friendly and fits on a single page of A4 paper in size 14 font. All aspects of treatment burden identified in a comprehensive evidence-based framework are included in the questionnaire. In comparison, the most comprehensive existing questionnaire, the PETS questionnaire,[8] includes 48 questions and is time-consuming to complete, and several of the other existing questionnaires focus on only some aspects of treatment burden.[6][7] Preliminary assessment of responsiveness found that, as expected, a positive change in both quality of life (EQ-5D-5L)[14] score and patient-centred care (PACIC)[16] score between baseline and 9-month follow-up was associated with a reduction in treatment burden (MTBQ) score. Of the other relevant PROMs, only the HCTD has been assessed for responsiveness,[6] but the HCTD addresses fewer topics and has a narrower range of response options, possibly contributing to its greater problems with skewness and floor effects.

A limitation of this study is that the MTBQ was developed using a framework of treatment burden developed from qualitative study in the USA.[1] However, apart from the issue of paying for care, we felt that other domains of treatment burden were likely to be generalisable, and we wanted to develop a measure that covered generic issues that would be relevant in a range of settings rather than specific to one healthcare system. Our measure was also informed by qualitative papers from different countries (including the UK) to ensure we included the important concepts.[10–12] In cognitive interviews, participants with multimorbidity felt that the questionnaire captured the range of factors that contribute to treatment burden.

A further limitation is that the participants of this study were recruited into a trial, which creates potential for selection bias and may limit generalisability. However, the trial participants had similar characteristics to those invited but declining participation in respect of age, gender, number and type of long-term conditions (data will be reported with the 3D trial results). Almost all the participants of this study were white British and further work is planned to validate the questionnaire in other populations. We found high floor effects with 22% of participants scoring a global MTBQ score of 0. All of the other treatment burden measures also show similarly high floor effects.[4–8] One explanation for this is a 'response shift', whereby patients adapt their everyday life so that looking after their health conditions becomes more acceptable to them over time and causes less perceived burden.[23] The following are the implications of positively skewed treatment burden scores and high floor effects: first, this can make it difficult to detect change (ie, it is not possible to improve from a treatment burden score of 0); and second, mean treatment burden scores should be interpreted with caution. Preliminary analysis of responsiveness, however, has shown that changes in MTBQ score correlate as expected with changes in quality of life (EQ-5D-5L)[14] score and patient-centred care (PACIC)[16] over time. We recommend that, due to the skewness of global MTBQ scores, researchers should report the median and IQR rather than the mean and SD, and report the proportion of patients with high, medium, low or no treatment burden (MTBQ scores ≥22, 10–22, <10 and 0, respectively).

The MTBQ scale is a concise measure of treatment burden for patients with multimorbidity that has demonstrated good content validity, construct validity, internal consistency reliability and responsiveness. It is a useful research tool for assessing the impact of interventions on treatment burden for patients with multimorbidity. We anticipate the scale being used alongside other measures, such as disease burden, and that findings from the two measures will be related. The MTBQ could also be used in clinical practice to highlight problem areas for patients with multimorbidity, such as difficulties the patient may have with their medication or with making recommended lifestyle changes. Further work is needed to validate the MTBQ for use in a clinical setting.

**Acknowledgements** Appreciation is extended to members of the patient and public involvement group, known as the Patient Involvement in Primary Care Research (PIP-CaRe) group, who took part in the cognitive interviews. The PIP-CaRe group was formed for the purpose of the 3D Study and consists of people with two or more long-term conditions. We also thank all other members of the 3D research team and Professor Boyd for permission to use the Healthcare Task Difficulty questionnaire.

**Contributors** PD, M-SM and CS were responsible for study concept and design. PD, MM, DG and KC were involved in data extraction and analysis. PD drafted the manuscript. All authors critically reviewed the manuscript and approved the final version. All authors also had full access to all of the data (including statistical reports and tables) in the study and can take responsibility for the integrity of the data and the accuracy of the data analysis. PD is the guarantor. PD led this project under the supervision of CS. She designed the study, undertook a literature

review, developed the questionnaire, conducted and analysed the cognitive interviews, convened meetings with the patient and public involvement group, analysed the results, and drafted the paper. MM provided methodological expertise in assessing the psychometric properties of this new patient-reported outcome measure, including the approach to analysis and interpretation of the results. She critically appraised the paper and has approved the final version. M-SM provided methodological and practical expertise, and obtained ethical and governance approvals for this study. She critically appraised the paper and has approved the final version. KC acquired and cleaned the original data and produced the database used for analysis. She critically appraised the paper and has approved the final version. DG provided methodological expertise in analysing the responsiveness of the MTBQ and the interpretation of these results. She critically appraised the paper and has approved the final version. CS was Chief Investigator of the 3D Study, which formed the basis for this paper, and supervised PD in developing this questionnaire. He contributed to study design, analysis and interpretation. He critically appraised the paper and has approved the final version.

**Funding**  This work was funded by the National Institute for Health Research Health Services and Delivery Research Programme (project number 12/130/15).

**Disclaimer**  The views and opinions expressed therein are those of the authors and do not necessarily reflect those of the HS&DR Programme, NIHR, NHS or the Department of Health.

**Competing interests**  None declared.

**Patient consent**  Not required.

**Ethics approval**  The 3D Study was approved by South-West (Frenchay) NHS Research Ethics Committee (14/SW/0011).

**Provenance and peer review**  Not commissioned; externally peer reviewed.

**Data sharing statement**  Data will be available from the University of Bristol Research Data Storage Facility after the main results of the 3D trial have been published in 2018.

**ORCID iDs**
Polly Duncan http://orcid.org/0000-0002-2244-3254
Mei-See Man http://orcid.org/0000-0003-4948-5670
Katherine Chaplin http://orcid.org/0000-0003-1261-9938
Chris Salisbury http://orcid.org/0000-0002-4378-3960

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
