## [Reviewer comments · BMJ Open]

ARTICLE DETAILS

TITLE (PROVISIONAL)	Development and validation of the Multimorbidity Treatment Burden Questionnaire (MTBQ)
AUTHORS	Duncan, Polly; Murphy, Mairead; Man, Mei-See; Chaplin, Katherine; Gaunt, Daisy; Salisbury, Chris

VERSION 1 – REVIEW

REVIEWER	TRAN Viet Thi INSERM France
REVIEW RETURNED	24-Sep-2017

GENERAL COMMENTS	Thank you for letting me review this article on the development of a new instrument to measure the burden of treatment that complements existing tools and options. The study finds its strength in its sample size and by being one of the few studies that assesses responsiveness. The development of this instrument follows a classic approach and is well reported. Despite, I have some comments: Major comments: 1) The authors state in the introduction that the TBQ (in English) had a too complex wording. Did the authors test the TBQ in this or another study? Although it may be true, this is a rather strong affirmation. Besides having been tested online during the English adaptation study, the TBQ was firstly developed with participants recruited "face to face" in waiting rooms of general practitioners' clinics and in hospital departments with a population that may be closer to which of the present study (although not all participants were multimorbid). As there is an evident overlap in the content of the final items of the MTBQ and of the TBQ (besides wording), I'm surprised why did the authors feel like it was necessary to undergo the full development of a new measure (and as I understand it, simply based on their impression on the wording of an existing instrument)? 2) As this study was nested in the trial, the authors used for construct validity the other outcomes collected in the trial. However, it would have been interesting for the validation of the present instrument to assess the agreement between the different measures of the burden of treatment (TBQ, PETS, MTBQ). Indeed, construct validity is usually used for measures that don't have any counterpart: in the present case, a criterion validity with the PETS or TBQ would have been more indicated (and it is discussed in the ISOQOL article appended to the present manuscript). 3) In this study, authors considered as responsiveness the ability of their developed instrument to change alongside with QOL as measured with the EQ-5D. This is, for me, a strength of the study.
--

	However, it opens a discussion about the hypotheses tested: does a change in QOL truly reflect the change of burden of treatment and demonstrates the responsiveness of the instrument? I would have been more interested to see how the MTBQ could vary with varying treatment strategies in the time period. But, this may represent another work. 4) I'm not completely sure how missing data were handled 5) I'm not completely sure about why, in the results, the authors report the number of people who answered at least "half of the MTBQ questions". Later, the authors state that only few people had missing values. How many participants were included in the 3D study, how many completed all items from the MTBQ? If some participants completed only half of the questionnaire's items, how did you manage missing values? (Very) Minor comments: The English version of the TBQ was not tested with volunteers from an internet forum but with patients from PatientsLikeMe on the ORE platform
--	--

REVIEWER	Katie Gallacher University of Glasgow Scotland
REVIEW RETURNED	14-Dec-2017

GENERAL COMMENTS	This is a well written paper that describes the development and validation of a scale that measures treatment burden in multimorbid patients. The scale has 10 items which seems user-friendly. As the author's state, the use of a trial population is a limitation. The paper mainly focusses on the psychometric analyses and I recommend that more information is given on the measure development. I ask that the authors address the following points: 1) Page 3: Please add PACIC to the abbreviations. 2) Page 6, line 17: The authors state that a review of the literature informed the measure development yet little information is given about how they conducted this review and what measures were found. Detail should be added to the methods and results sections. 3) Page 6, line 18: It is stated that existing PROMs were reviewed against an existing framework (Eton et al). More information is required here. Which measures were used for this? A brief description of the framework would be helpful as the reader may not be familiar with Eton's paper. 4) Page 6, line 18: The framework used by Eton et al was developed in a US population that included an ethnically and socioeconomically different group. This should be mentioned in the limitations. This study would have benefited from a qualitative study in a UK patient population. 5) Page 6, line 19: More information is needed here. Was there an initial focus group with all 14 members? If so, more detail is needed about this focus group. How was it conducted? What was asked of the group? What was their feedback? Detail should be added to the methods and results sections. 6) Page 6, line 23: Why were only two rounds carried out? Usually this is guided by data saturation i.e. no further changes to the items needed. Two rounds seems quite few, but this may have been due to the small numbers of questions. Authors should please
---

	clarify this. 7) Page 6, line 48: More information about why the HCTD questionnaire was chosen. 8) Page 7, line 24: This section seems to repeat a section on the previous page, word count could be cut if duplication is avoided. 9) Page 11, line 13: information about the participants' socioeconomic status and ethnicity should be given here. 10) Page 12, line 12: The authors need to justify why they did not amalgamate the questions that had high inter-item correlation. This may have been for clinical / pragmatic reasons but a reason should be given. 11) Page 12, line 25: It is stated that all but one of the participants felt that the important areas of treatment burden were covered. More information is needed here. What were the burdens they felt were missing? Why were the items in the measure not then amended? 12) Page 15, line 8: socio-economic deprivation should be mentioned here when comparing those invited to those who took part. 13) Page 19, line 39: There is a duplicated reference (Reeve et al). 14) Page 20, table 1: a) There is a typo at CV disease under 'main study'. B) More explanation is needed about the deprivation score – a mean score is unlikely to be meaningful to the reader. Quartiles/quintiles would be more helpful. C) This table is difficult to read – particularly the bottom section 'outcome measures'. I suggest taking out this section and either putting it in another table or in the main text. It is interesting that number of self-reported comorbidities is much higher than the number taken from the GP records (discussion of this probably outwith the realms of this paper though). 15) Page 24, line 30: there is no convincing relationship between deprivation and treatment burden. This is interesting and should be mentioned in the main text. It may be explained an over representation of affluent participants. There is a general lack of mention of socio-economic deprivation that needs addressed.
--	--

REVIEWER	Orla C. Sheehan Assistant Professor, Center on Aging and Health, Division of Geriatric Medicine and Gerontology, Johns Hopkins University, USA
REVIEW RETURNED	18-Dec-2017

GENERAL COMMENTS	A simple tool to evaluate treatment burden in patients with multiple chronic conditions is badly needed and I applaud your work. I would advise explaining that the comparator you chose in most of your discussion and in table 3, the difficulty with health care tasks scale (HCTD), was designed to measure one aspect of treatment burden only and therefore comparing it to your measure designed to capture all aspects of treatment burden is not a fair comparison. The author of the HCTD paper states "Difficulty with health care tasks (HCTD) (e.g. difficulty planning medication schedule) is one aspect of treatment burden." (Med Care 2014, Boyd). Please temper your comparisons or limit your comparisons to the aspects of the MTBQ captured by the HCTD.
--

REVIEWER	Norma O'Flynn National Guideline Centre, Royal College of Physicians, 11 St, Andrews Place, London NW1 4LE, UK
-----------------	---

REVIEW RETURNED	21-Dec-2017
GENERAL COMMENTS	This paper presents an important validation of a PROM for treatment burden in particular the demonstration of change over time and guidance on interpretability. The changes I would suggest are minor and depend in part on how the measure is framed. It is currently framed as a multimorbidity measure for use in research with a sentence at the end saying it could be used clinically. The development within a specific study has strengths and limitations- it has allowed aspects such as change over time to be examined but limits the population to those included in the study. The inclusion criteria, i.e. 3 or more longterm conditions, results in this validation being restricted to this population and limits the population to those who fulfill this particular definition of multimorbidity. Less defined but common conditions such as pain have not been included and I did not see any measure of number of medicines being taken. The paper states that further validation in a more ethnically diverse population is planned. The wider use of the measure in research clinically, would depend on its properties in assessing treatment burden in a less restricted populations and if this is being suggested validation is required in these populations as well.

VERSION 1 – AUTHOR RESPONSE

Reviewer 1.

Response to initial comment:

We agree, a strength of this study is that it has been validated in a large sample of older people with three or more long-term conditions. As noted, it is also one of the few studies that assesses responsiveness. This is particularly important for research trials wanting to assess the effect of an intervention on treatment burden.

We have developed and validated the scale using a commonly accepted approach. For ease of reading, we have structured the analysis plan and results under the subheadings of the minimum standards for a patient reported outcome measure set out by set out by the International Society for Quality of Life Research (ISOQOL).

Response to major comments:

1) The scale was developed and validated as part of the 3D Study, a randomised controlled trial to improve the management of multimorbidity (three or more long term conditions) within primary care. In order to select outcome measures for the trial, the 3D Study research team reviewed existing measures of treatment burden with the Patient and Public Involvement (PPI) group and they unanimously reached the view that the wording of the TBQ was likely to be too complex for the participants of the 3D Study (mean age 71 years; in a range of settings including deprived areas; some patients with low literacy levels).

Since the TBQ and MTBQ both set out to measure treatment burden it seems inevitable that there will be overlap in the content of the questions. We have described in the methods section of the paper how we developed the questionnaire. This included reviewing the existing measures of treatment burden against a framework of treatment burden described by Eton et al. We concluded that there was no existing measure which was brief, readily understandable by elderly patients with

multimorbidity, and covered all of the necessary domains. This was why we undertook the study. We believe our new measure will be valuable particularly in other trials which also need a brief measure of treatment burden.

The issue of brevity is important. Such trials of multimorbidity interventions will often require a number of outcome measures and patients will be asked to complete a battery of different measures within one questionnaire at baseline and follow-up. It is a major advantage if the measure takes up no more than one page within a questionnaire booklet.

2) The 3D Study research team chose to include the Healthcare Task Difficulty (HCTD) questionnaire as a comparator questionnaire because we felt that the wording was simple and therefore suitable for the older and less literate participants of the study. It also covered some (but not all) of the concepts we wanted to measure. At the time we conducted this work the PETS had not been published, and in any case the fact that it contains 48 items would have made it too long to use as one of a number of secondary outcome measures in a trial.

To assess criterion validity, we tested the pre-specified hypothesis of a positive association between global MTBQ score and global HCTD.

We have added the following to justify why we chose to use the HCTD as a comparator:

“A key reason for choosing this measure was the simple wording and brevity. This was felt to be important because many of the participants of the 3D study were older people and some had low literacy levels.” (page 7)

We agree that the next step would be to carry out a direct comparison between participants' responses to the MTQB, TBQ, and PETS, including asking participants to rate how easy the questionnaires were to complete. We are planning to undertake such a study. Perhaps this could be a collaborative project?

3) We agree that assessing responsiveness to change is a strength of this study. The EQ-5D-5L focuses on mobility, self-care, usual activities, pain/discomfort and anxiety/depression. It seems reasonable that participants with improved quality of life score over time (e.g. improved mobility, reduced pain or improved ability to look after themselves), might report reduced perceived treatment burden over time. We also hypothesised an inverse association between change in patient-centred care (PACIC) score and treatment burden score. We have been quite clear in the discussion chapter, describing the responsiveness results as a 'preliminary assessment of responsiveness' and we agree that further work could be done to assess responsiveness more comprehensively.

4) Handling of missing data is described in the methods section under the subheading 'Construct validity':

“Each question was scored as follows: zero (not difficult/ does not apply), one (a little difficult), two (quite difficult), three (very difficult), four (extremely difficult). Participants were excluded if more than 50% of their responses were missing. To calculate a global score, each participant's average score was calculated from the questions answered and multiplied by 25 to give a score from 0-100.” (page 8)

This 'half-scale rule' is a common approach to managing missing data in other self-report questionnaires e.g. the SF36.

5) We chose to report the number of participants who had completed at least half of the questions because a global MTBQ score could be calculated for these participants (see comment above re:

handling of missing data). For participants who completed half of the questions, we calculated an average score and multiplied by 25 to give a score between 0 and 100.

We have slightly changed the wording of the 'Participant Characteristics' at the beginning of the results chapter. This now reads:

"143 adults participated in the pilot study. From 1546 participants in the main 3D study who completed the main baseline questionnaire we were able to calculate a MTBQ score for 1524 (99%) individuals who completed at least half of the baseline MTBQ questions. At nine-month follow up, 1356 returned the questionnaire and a MTBQ score could be calculated for 1299 (96%)." (page 10)

Response to minor comments:

We have amended the introduction as follows:

"...volunteers recruited from the 'Patients like me' website (mean age 51 years, 78% with college education), not all of whom had multimorbidity." (page 5)

Reviewer 2.

Response to initial comments:

We set out to develop and validate a concise user-friendly scale of treatment measure for patients of multimorbidity.

Response to comment re: using a trial population:

Under the 'Article Summary' section of the paper, we have clearly stated that participants were recruited into a trial and that this may limit generalisability of the study. We have also included this limitation in the discussion chapter. However the trial itself had very broad inclusion criteria, included 33 practices from a range of settings, and participants were representative of all eligible patients with multimorbidity within participating practices. Therefore the patient sample in this study may well be at least as representative of patients with multimorbidity as those who have volunteered to be included in other studies to develop other questionnaires such as the TBQ and the PETS.

Response to request for more information about the development of the measure:

1) PACIC has been added to the list of abbreviations (page 3)

2) We have amended the methods chapter to include more information about our review of the literature and we have outlined the existing PROMs that were identified. We provide some further information about the existing PROMs in the introduction and discussion chapters.

The 'Development of the questionnaire' paragraph now reads:

"We reviewed the literature on the concept and measurement of treatment burden in multimorbidity using Pubmed in July 2014. We identified a number of relevant qualitative studies¹⁰⁻¹² and three relevant existing PROMs that were not specific to a particular medical condition. These were the Treatment Burden Questionnaire (TBQ),⁴ the Multimorbidity Illness Perceptions Scale (MULTIPLES)⁷ and the Healthcare Task Difficulty (HCTD) questionnaire.⁶ A further measure, the PETS scale, was published later.⁸ We identified relevant domains for the PROM by reviewing the three existing PROMs against a framework of treatment burden which had been developed following qualitative interviews and focus groups.¹" (page 6)

3) An outline of the existing framework is included in the results chapter under the heading 'Conceptual framework':

"The framework developed by Eton et al,¹ describes three major themes of treatment burden: the work required to look after one's health (e.g. self-monitoring, making lifestyle changes); tools and strategies patients use to reduce their treatment burden (e.g. organising medication); and factors that increase burden (e.g. poor continuity of care). We mapped the three existing treatment burden questionnaires against this framework, and discussed this with the PPI group who felt that all of the domains of treatment burden identified in the literature should be included in the PROM." (page 10)

4) We have added the following to the discussion chapter:

"A limitation of this study is that the MTBQ was developed using a framework of treatment burden developed from qualitative study in the United States.¹ However, apart from the issue of paying for care, we felt that other domains of treatment burden were likely to be generalisable and we wanted to develop a measure that covered generic issues which would be relevant in a range of settings rather than specific to one health care system. Our measure was also informed by qualitative papers from different countries (including the UK) to ensure we included the important concepts.¹⁰⁻¹² In cognitive interviews, participants with multimorbidity felt that the questionnaire captured the range of factors that contribute to treatment burden."

5) Eight of the 14 members of the PPI group for the 3D study attended a meeting to discuss the concept and measurement of treatment burden in multimorbidity. We showed them existing measures and asked them to comment on the range of topics they covered and their wording and presentation. We have changed the methods chapter as follows:

"We then sought the views from a Patient and Public Involvement (PPI) group of eight patients with multimorbidity formed for the purpose of the 3D Study, discussing the concept of treatment burden, the existing measures, the treatment burden framework and the domains of treatment burden to be included in the questionnaire." (page 6)

We have also included a section in the results chapter under 'Conceptual framework' outlining the feedback from the PPI group. This reads as follows:

"We had initially considered excluding questions about costs since health care is mostly free under the National Health Service, but our PPI group argued that they still experienced additional costs from managing illness so this domain was retained in the first draft." (page 10)

6) We have added the following to clarify why we chose to only undertake two rounds of cognitive interviews:

"The second round of cognitive interviews led to only minor changes to the questionnaire with no new insights emerging." (page 6)

7) We chose the HCTD as the comparator in the pilot study because we had concluded, after discussion with the PPI group and research team, that we would use the HCTD as the most suitable alternative measure of treatment burden for the 3D trial if our new MTBQ measure did not prove to be valid and reliable.

We have added the following to the paper:

"A key reason for choosing this measure was the simple wording and brevity. This was felt to be important because many of the participants of the 3D study were older people and some had low literacy levels." (page 7)

8) We have deleted this paragraph and replaced it with the following:

"See 'Development of the questionnaire'." (page 7)

9) We have added that 99% of participants were white British and described the deprivation levels. This now reads:

“The participants were mostly elderly (mean age 71 years for the main study), fully retired from work, had left school aged 16 years or younger and 99% were white British (Table 1). Around two-thirds of participants from England lived in areas of low deprivation (lower or middle lower quartiles), whereas almost two thirds of participants from Scotland lived in areas of high deprivation (middle upper or upper quartiles).” (page 10)

10) The choice of items for questionnaires always involvements an element of judgement, guided rather than dictated by statistical considerations. We have now justified why we decided not to amalgamate the questions with high inter-item correlation. This reads as follows:

“A decision was made to include questions 1 and 2, and 6 and 7 despite the high inter-item correlation coefficients because it was felt these questions were about different aspects of treatment burden.” (page 11)

11) We have added more information as suggested:

“One participant felt that accessing health care outside of usual GP opening hours caused significant treatment burden for him. In response to this, we added a question about difficulty getting health care in the evenings and weekends (question 10). The remaining participants commented that the important areas of treatment burden were covered by the questionnaire.” (page 11)

We have also amended the methods chapter to highlight that a question was added as a result of the cognitive interviews. (page 6)

12) Unfortunately we do not have the postcodes for participants who were invited to take part in the study but declined, as they did not consent to the research team having access to this information. The postcodes were used to calculate the Index of Multiple Deprivation (IMD) scores and so it was not possible to compare IMD scores between those who took part and those who declined to take part.

13) Duplicate reference removed.

14) a) Typo corrected (page 19)

14) b) We have now calculated the proportion of participants living in areas of low, lower middle, upper middle and upper quartiles of deprivation, using the England IMD 2010 data and Scotland IMD 2012 data as reference.

We have also changed Table 1 to include deprivation score quartiles. We have changed the footnote for Table 1 as follows:

“Individual Index of Multiple Deprivation (IMD) score, 2010, for England, and Scottish Index of Multiple Deprivation (SIMD) score, 2012, for Scotland, based on participants postcodes. The lower quartile is the least deprived and the upper quartile is the most deprived” (18)

We have added a sentence about deprivation level to the ‘Participant Characteristics’:

“Around two-thirds of participants from England lived in areas of low deprivation (first or second quartiles), whereas almost two thirds of participants from Scotland lived in areas of high deprivation (third or fourth quartiles).” (page 10)

14) c) We are limited by the need not to exceed 5 tables. We have changed the label for this section of the table to ‘Baseline scores of outcome measures’ (page 18)

The reason for the higher number of self-reported conditions is because patients selected from a longer list of 27 chronic conditions, whereas the inclusion criteria were based on patients who had 3 or more from a list of 11 major groups of conditions, based on the medical records. We have changed the footnote about the self-reported conditions. This now reads:

“Number of self-reported conditions from a list of 27 conditions itemised in the Bayliss scale”. (page 18)

15) We have added a sentence to the ‘Interpretability of scores’ paragraph in the results chapter, as follows:

“There was no convincing association between deprivation score and treatment burden score.” (12)

We have also commented on this lack of association in the discussion chapter, as follows:

“There was no association between deprivation level and treatment burden score. One might expect that people from more deprived areas might have fewer support networks and resources and so would experience higher treatment burden. Alternatively, one could argue that participants from more deprived areas might be more accepting of how looking after their health impacts on their day to day life and so report lower treatment burden.” (page 13)

Reviewer 3.

Response to initial comments:

We agree that there is a need for a simple measure of treatment burden for patients with multimorbidity. This is illustrated by the considerable interest we have had in our questionnaire. The MTBQ has been translated into Danish for use in the Central Denmark Health Survey, a cross-sectional survey of approximately 54,000 participants. It is also being used in a number of trials.

Response to comment re: using the HCTD as a comparator:

The 3D Study research team chose to include the Healthcare Task Difficulty (HCTD) questionnaire as a comparator questionnaire because we felt that it was the most suitable existing measure for the older and less literate participants of the study. To clarify our reasons for choosing the HCTD as a comparative measure, we have added the following:

“A key reason for choosing this measure was the simple wording. This was felt to be important because many of the participants of the 3D study were older people and some had low literacy levels.” (page 7)

We have acknowledged both in the introduction and the discussion chapter that a limitation of the HCTD is that it focuses on one aspect of treatment burden. This is also acknowledged by the authors of the HCTD. However we would argue that, since both scales are measuring some aspects of treatment burden, it is reasonable to hypothesise a positive association between global HCTD score and global MTBQ score.

Reviewer 4.

Response to initial comment:

We agree that the assessment of both responsiveness and interpretability are strengths of this study.

Response to the comment re: framing of the measure:

We agree that a strength of validating the MTBQ within a trial setting is that we were able to assess responsiveness to change. We have also acknowledged that this may limit the generalisability of the study and that further work is planned to validate the measure in other populations.

The inclusion criterion for this study (i.e. patients with multiple long term conditions) is the mostly widely used approach for defining patients with multimorbidity. Some other treatment burden measures have been developed in populations which do not fulfil this definition of multimorbidity, and often with a much smaller number of participants. We would therefore argue that the population included in the study is less restricted than some other studies and this is a strength rather than a limitation of our study.

We have amended the final paragraph, stating that the measure could potentially be used in a clinical setting for patients with multimorbidity but further work would be needed to validate the measure in this setting. The amended final paragraph reads:

“The MTBQ could also be used in clinical practice to highlight problem areas for patients with multimorbidity, such as difficulties the patient may have with their medication or with making recommended lifestyle changes. Further work is needed to validate the MTBQ for use in a clinical setting.” (page 15)

VERSION 2 – REVIEW

REVIEWER	Katie Gallacher University of Glasgow Scotland
REVIEW RETURNED	23-Jan-2018

GENERAL COMMENTS	The authors have addressed all of my comments and all comments from the other reviewers. The paper has been improved following amendments. I have no further changes to request
---

REVIEWER	Norma O'Flynn Royal College of Physicians, St Andrews Place, London, UK
REVIEW RETURNED	26-Jan-2018

GENERAL COMMENTS	The authors have addressed the issues raised in the review process to my satisfaction.
--

REVIEWER	Orla Sheehan Assistant Professor Center on Aging and Health Division of Geriatric Medicine and Gerontology Johns Hopkins University United States
REVIEW RETURNED	26-Jan-2018

GENERAL COMMENTS	Thank you for your thoughtful responses to all the reviewer's comments. I believe that the revised manuscript is much improved.
---